# Capillary Networks for Bio-Artificial Three-Dimensional Tissues Fabricated Using Cell Sheet Based Tissue Engineering

**DOI:** 10.3390/ijms22010092

**Published:** 2020-12-23

**Authors:** Hidekazu Sekine, Teruo Okano

**Affiliations:** 1Institute of Advanced Biomedical Engineering and Science, Tokyo Women’s Medical University, Tokyo 162-8666, Japan; 2Center for Advanced Medical and Life Science, Tokyo Women’s Medical University, Tokyo 162-8666, Japan; tokano@twmu.ac.jp; 3Cell Sheet Tissue Engineering Center (CSTEC), Department of Pharmaceutics & Pharmaceutical Chemistry, School of Medicine and College of Pharmacy, University of Utah, Salt Lake City, UT 84112, USA

**Keywords:** regenerative medicine, tissue engineering, cell sheet technology, vascularization, vascular bed, bioreactor, tissue culture

## Abstract

One of the most important challenges facing researchers in the field of regenerative medicine is to develop methods to introduce vascular networks into bioengineered tissues. Although cell scaffolds that slowly release angiogenic factors can promote post-transplantation angiogenesis, they cannot be used to construct thick tissues because of the time required for sufficient vascular network formation. Recently, the co-culture of graft tissue with vascular cells before transplantation has attracted attention as a way of promoting capillary angiogenesis. Although the co-cultured vascular cells can directly contribute to blood vessel formation within the tissue, a key objective that needs to be met is the construction of a continuous circulatory structure. Previously described strategies to reconstruct blood vessels include the culture of endothelial cells in a scaffold that contains microchannels or within the original vascular framework after decellularization of an entire organ. The technique, as developed by authors, involves the progressive stacking of three-layered cell sheets onto a vascular bed to induce the formation of a capillary network within the cell sheets. This approach enables the construction of thick, functional tissue of high cell density that can be transplanted by anastomosing its artery and vein (provided by the vascular bed) with host blood vessels.

## 1. Introduction

Regenerative medicine has received considerable attention as a new approach to the treatment of intractable diseases that cannot be cured using current medical and surgical strategies, and this technique is expected to replace organ transplantation in the future. Cell infusion therapy is a form of regenerative medicine that has already been applied in the clinical setting and involves the injection of a cell suspension derived from a patient or other source into failing tissue. However, the limitations of cell infusion therapy have driven the development of tissue engineering, which represents the next step in regenerative medicine. Current tissue engineering strategies are based on the seeding of cells onto biodegradable polymer scaffolds or decellularized scaffolds, and these methods are well suited to the generation of tissues with low cell densities and low vascular requirements such as bone, cartilage, and skin [1]. Tissue engineering techniques overcome some of the disadvantages of cell infusion therapy such as cellular necrosis, poor cell retention at the target tissue, and unsuitability for the treatment of defects associated with congenital diseases. However, since conventional tissue engineering technologies rely on simple diffusion to deliver oxygen/nutrients and remove waste products, tissues generated with these approaches are limited in terms of their thickness and functionality. The construction of tissues with higher cell densities, more complex structures, and higher vascular requirements (such as heart, liver, and kidney) will require the development of innovative techniques to achieve functional vascularization of the bioengineered graft. This review describes some of the methods that can be used to construct vascular networks within bioengineered three-dimensional (3D) organs, with a focus on research into the generation of myocardial tissue.

## 2. Overview

### 2.1. Scaffold-Based Tissue Engineering

Tissue engineering is a field of study that emerged from a fusion of medicine and engineering, and it combines aspects of cell biology, physical chemistry, and materials engineering to create solutions for the building or regeneration of tissue and organ structures. Initially, it was thought that tissue construction would require cells, an extracellular matrix as a scaffold for the cells, and cytokines to promote cell differentiation and proliferation. Therefore, in early studies, cells were seeded onto a biodegradable polymer scaffold made from polylactic acid and its copolymers, cultured, and then transplanted into the body. The scaffold would be gently degraded and absorbed in vivo to be replaced by an extracellular matrix produced by the cells, which were expected to self-assemble [2]. A major advantage of tissue engineering is that it overcomes a major drawback of cell infusion therapy, namely cell loss and necrosis due to the lack of a scaffold for the cells to attach to. Furthermore, tissue engineering can be used to treat defective sites such as those occurring in congenital diseases, which is something that cannot be achieved with cell infusion or cytokine administration therapies [3]. 

Porous sponges made of gelatin, alginate, or polylactic acid have been the most commonly used scaffolds for cell seeding in myocardial tissue engineering (Figure 1A) [4,5,6]. For example, Li et al. seeded fetal rat cardiomyocytes into biodegradable mesh gelatin and transplanted them onto myocardial scar tissue in a cryoinjured rat heart [5]. Leor et al. seeded fetal rat cardiomyocytes onto an alginate-based porous scaffold and implanted them onto the hearts of rats with experimental myocardial infarction; although grafting did not improve left ventricular contractility, it did prevent left ventricular enlargement secondary to myocardial remodeling [6].

Collagen has also been used as a scaffold for cardiac tissue engineering. Zimmermann et al. reported that 3D myocardial tissue could be made to a chosen shape by culturing a mixture of neonatal rat cardiomyocytes and collagen solution in a silicone mold and stretching of the tissue in vitro was found to improve the orientation, size, and maturity of the cardiomyocytes (Figure 1B) [7]. Additional experiments in an animal model of myocardial infarction revealed that the transplanted tissue became electrically connected to normal myocardium without the induction of arrhythmias, resulting in an improvement in left ventricular contractility and inhibition of left ventricular enlargement [7].

Alternative methodologies have also been described. An intermediate approach between cell infusion therapy and tissue engineering, in which cells were mixed with a solution of fibrin or collagen and then injected into failing myocardium, was reported to reduce the cell loss associated with the infusion of cell suspensions (Figure 1C) [8]. In addition, an attempt has been made to generate a functional organ by decellularizing a heart while retaining its vascular architecture and then re-seeding it with suspensions of cardiac or endothelial cells (Figure 1D) [9].

Recently, a technique has been developed that uses a 3D printer to precisely place cells and thereby assemble 3D tissues in a bottom-up fashion (Figure 1E) [10]. 3D printing uses computer-designed digital data as a blueprint for modeling, allowing the size and shape of each piece of tissue to be uniquely determined. This method enables tissue pieces to be generated in a short time and allows the internal structure of the object to be easily designed, which is difficult to do with casting processes. However, many restrictions apply to this technique because the living cells have to be embedded in the ink, such as the need to keep the temperature at around 37 °C at the time of printing and the requirement of an aqueous solution with a neutral pH and physiological osmotic pressure. In addition, bio-ink must be formulated to be suitable for extrusion by the printer, and there are issues regarding its mechanical strength as a scaffold for cells.

### 2.2. Cell Sheet-Based Tissue Engineering

One limitation of using cell scaffolds for tissue construction is that it is difficult to seed a sufficient number of cells evenly within the scaffold, which results in tissues with few cellular components and a large amount of connective tissue. Thus, although scaffold-based methods might be suitable for the construction of simple tissues with relatively low cellularity such as heart valves and cartilage, new technologies are needed to generate cell-dense, structurally complex, and highly functional tissues such as heart, kidney, liver, and lung. With this in mind, a new technology called cell sheet engineering was developed, which has the potential to generate highly functional tissues without the use of a scaffold. Cell sheet engineering relies on a technology that modifies the surface of a culture dish so that the adhesion and detachment of cells can be controlled through changes in temperature. Poly(N-isopropylacrylamide), a temperature-responsive polymer with a lower critical solution temperature of 32 °C, is covalently fixed to the surface of a culture dish with an electron beam. Cells are able to adhere to the polymer surface at 37 °C because poly(N-isopropylacrylamide) is weakly hydrophobic at this temperature. However, reducing the temperature below 32 °C causes poly(N-isopropylacrylamide) to become hydrophilic and the cells to detach [11,12]. Conventional approaches rely on trypsin or other proteases to harvest cells from a culture dish, but proteases inevitably cause the degradation of many different membrane proteins and not just those anchoring the cells to the dish surface. By contrast, the use of a temperature-responsive culture dish permits cells to be harvested as a sheet without damage to cell-cell junctions or the extracellular matrix (Figure 2) [13]. Furthermore, 3D tissue can be constructed by stacking cell sheets on top of each other (Figure 1F). Additionally, tissues built from cell sheets consist only of the relevant cell type(s) and a small quantity of an extracellular matrix produced by the cells, which circumvents some of the issues related to the use of biodegradable scaffolds [14].

When two cell sheets prepared from neonatal rat cardiomyocytes using temperature-responsive culture dishes were stacked in vitro, morphological and electrical connections formed between the stacked cardiomyocyte sheets within a few minutes, indicating that this method has the potential to generate synchronized, autonomously beating tissue [15,16]. After transplantation into the subcutaneous tissue of a rat, the stacked cardiomyocyte sheets exhibited spontaneous beating and generated electrical potentials that were distinct from those of the host heart. Furthermore, the transplanted tissue resembled native myocardial tissue and contained capillary networks, cylindrical myocardial structures, gap junctions, and desmosomes [15]. Notably, the transplanted myocardial graft remained viable and maintained its spontaneous beating for the ~2-year lifespan of the rat [17].

Experiments using an animal model of heart failure revealed that the transplantation of stacked cardiomyocyte sheets resulted in the formation of gap junctions and electrical connections between the graft tissue and host myocardium [18]. In addition, the rate of cell engraftment after transplantation was shown to be much higher for cell sheets than for cell suspensions [19]. Moreover, it was found that the co-culture of vascular endothelial cells with cardiomyocytes led to the creation of a network of vascular endothelial cells that directly contributed to angiogenesis and accelerated the recovery of ischemic heart function after transplantation of the cardiomyocyte graft [20].

### 2.3. Scalable Assembly Techniques to Create Vascularized Tissue

#### 2.3.1. In Vitro Approaches

Tissues constructed by conventional tissue engineering techniques do not have a functional vascular network and so suffer from a lack of oxygen and nutrients and an accumulation of waste products. Therefore, a major challenge facing researchers in the field of tissue engineering is to develop a method of introducing vascular networks into constructed tissues. The initial research in this area relied on angiogenesis from the host side to generate a vascular network in the grafted tissue after transplantation. In fact, many of the regenerative therapies currently being used in clinical practice rely on the host-side vasculature for the supply of oxygen and nutrients and removal of waste products. A widely used technique to promote angiogenesis is the slow release of angiogenic factors (such as vascular endothelial growth factor, basic fibroblast growth factor, and platelet-derived growth factor-BB) from the cellular scaffold [21]. Although this method can promote angiogenesis in the early stages of transplantation, the time required for the formation of a vascular network precludes its use in the construction of thick tissue because ischemic necrosis of the interior of the tissue occurs before a sufficient vascular network is generated. One approach that has received recent attention as a way of promoting the generation of a vascular network is to co-culture vascular cells with the target cells before transplantation. A technique to produce cell sheets containing endothelial cell networks by co-culturing vascular endothelial cells and cardiomyocytes in an appropriate ratio has been developed, enabling the faster generation of a vascular network after transplantation [20,22,23]. However, although co-cultured vascular cells directly contribute to the formation of tubular structures that function as blood vessels, they do not form continuous tubular structures in 2D culture, and the average migration rate of vascular endothelial cells in vitro is slow, namely less than 5 μm/hour [24]. Therefore, new methods are needed to generate 3D tissue in vitro. One strategy being actively pursued is the development of a microfluidic system to introduce artificial capillaries that are perfused with fluid (Figure 3A) [25]. Other techniques for constructing a vascular network include the use of microfabrication processes to create a scaffold that mimics a 3D microcirculatory structure (an “angiotip”), the lumen of which is seeded with vascular endothelial cells that are then subjected to perfusion culture (Figure 3B) [26]. In addition, attempts have been made to create perfusable tissues by decellularizing tissues or organs, seeding vascular endothelial cells within the original blood vessels, and performing perfusion culture to regenerate a vascular network (Figure 3C) [9,27,28]. Decellularized tissue scaffolds may be ideal templates for cell seeding because anatomically, they provide a 3D vascular architecture.

#### 2.3.2. In Vivo Prevascularization Approaches

Another strategy for generating capillaries within engineered tissue is the in vivo revascularization approach (also called tissue prefabrication), which uses the body as a natural bioreactor (Figure 3D) [29]. The simplest prevascularization approach is the subcutaneous pocket method, in which the engineered tissue is implanted onto easily accessible and well-vascularized host tissue to induce the growth of host microvessels into the graft. The flap method involves the implantation of engineered tissue into a muscle flap, waiting for a sufficient vascular network to grow into the graft, and then surgically anastomosing the entire flap (including the transplanted tissue and vascular pedicle) with host vessels [30]. In the artery-vein (A–V) loop method, an anastomosis is made between a large artery and vein to incorporate the A–V loop and fabricated tissue in a closed chamber. Over time, blood vessels sprout from the loop, and angiogenesis occurs in the fabricated tissue. A microvascular network is formed after a few weeks, allowing the tissue construct to be transplanted by vascular anastomosis with host vessels as for the flap method [31,32]. The flap and A–V loop methods have the great advantage that the entire structure is perfused immediately after transplantation because the vascular pedicle is directly connected to local vessels during microsurgery. However, the disadvantages are the requirement for two separate surgical procedures (transplantation of the fabricated tissue to the site of vascularization and subsequent transplantation of the entire structure to the final site) and the need to remove host blood vessels from the initial transplantation site, which results in significant tissue loss at the donor site.

#### 2.3.3. Cell Sheet-Based Vascularization Approaches

To address the limitations associated with the prevascularization approaches described above, cell sheet-based technology was developed that enables the construction of functional 3D artificial tissues that have a high cell density and substantial thickness. The thickness of the tissue that can be fabricated by conventional tissue engineering is constrained by limitations in the diffusion of oxygen and nutrients. Therefore, a method was developed to create thick and functional myocardial tissue by transplanting multiple cell sheets onto host tissue in a stepwise manner, leaving sufficient intervals between transplantation to allow angiogenesis to generate capillaries within the graft. Vascular endothelial cells and cardiomyocytes were co-cultured on a temperature-responsive culture substrate that enabled cell adhesion to be controlled through changes in temperature. After the cells had been cultured to confluency, a drop in temperature was used to harvest the cardiomyocytes from the culture substrate in sheet form while maintaining the network structure of the vascular endothelial cells. Subsequently, co-cultured myocardial/endothelial cell sheets were transplanted in stages onto rat subcutaneous tissue, with enough time given between transplantations for sufficient angiogenesis to occur from the host capillaries (Figure 4A). It has been established that the maximum thickness of tissue that could be transplanted at one time was 80 μm, equivalent to three stacked cell sheets [33]. When two triple-layered cell sheets were transplanted at intervals of 1–2 days, it was also found that their beating became synchronized after one week, indicating that they had become electrically connected. Furthermore, the transplantation of a total of 10 triple-layered cell sheets at daily intervals enabled the construction of beat-synchronized myocardial tissue with a thickness of approximately 1 mm [33]. Therefore, it has been shown that this technique could be used to construct thick artificial 3D myocardial tissue with a functional network in vivo.

However, direct multistage transplantation of cardiomyocyte sheets onto failing myocardium is impractical because of the need for repeated and frequent open-heart surgery. Therefore, a new two-step method comprising of ectopic grafting to create a thick graft within the host subcutaneous tissue followed by re-transplantation to the desired target site was devised.

Since re-transplantation required the graft tissue to be supplied with an artery and vein, the host’s existing vascular system was used to provide blood vessels that could be anastomosed to vessels at the re-transplantation site (Figure 4B). Three rat cardiomyocyte sheets were stacked and transplanted onto branches of the rat femoral artery and vein (the superficial caudal epigastric artery and vein), and a new triple-layered cardiomyocyte sheet was transplanted 24 h later (after the induction of a capillary network in the first triple-layered sheet). Two weeks later, the transplanted myocardial tissue was dissected together with the host femoral artery and vein and re-transplanted into the neck to simulate implantation into the heart wall. During re-transplantation, microsurgery was used to anastomose the femoral artery with the carotid artery and the femoral vein with the jugular vein so that the thick myocardial tissue graft was provided with a blood supply (Figure 4C). Notably, macroscopic observations made two weeks after re-transplantation revealed that the graft continued to exhibit strong and synchronized pulsations, which indicated that it was functioning autonomously.

The overriding goal in cardiac tissue engineering is to create a functional myocardial chamber that can independently generate clinically relevant pressure from its own spontaneous contraction. Therefore, in addition to developing a cardiac patch that can be implanted directly onto a failing heart, there is also interest in designing organ-like tissues, such as tubular or spherical structures, that can serve as circulatory aids in patients with heart failure. Therefore, the stepwise cell sheet stacking method was to investigate whether thicker, functional, pump-like myocardial tissue could be produced using the mesentery as a vascular bed. Cell sheets were prepared by culturing neonatal rat heart cells on temperature-responsive dishes. After harvesting, three cell sheets were stacked, wrapped around a section of the endotracheal tube, and cultured to create cardiac tubes. In some experiments (single-step group), a single heart tube was implanted into the mesentery of each rat. In other experiments (double-step group), one heart tube was implanted into the mesentery, and a second heart tube was inserted into the original tube one day later after a sufficient number of new blood vessels had been created in the first transplanted tube. Spontaneous pulsations were observed in the cardiac tubes from the first week after transplantation, and the mean internal pressure at four weeks after transplantation was 1.8 ± 1.0 mmHg in the single-step group and 2.5 ± 0.3 mmHg in the double-step group, suggesting that increasing the number of cell sheet layers improved the functionality of the cardiac tube [34]. These findings demonstrate that it is feasible to use the mesentery as a vascular bed for the creation of functional cardiac tubes. Experiments to further develop this technology will include the fabrication of larger myocardial tubes and evaluation of their function after implantation into a host with anastomosis of the graft vessels with major vessels of the host.

Based on the results of prevascularization in an in vivo environment, an in vitro system was developed to bioengineer thick sections of functional tissue that could be perfused via blood vessels. This in vitro system consisted of a tissue perfusion bioreactor that incorporated biochemical stimuli to mimic the in vivo environment and a free vascular bed to impart capillaries into the cell sheets that were stacked on it (Figure 5). Specifically, a flap of thigh muscle containing well-developed blood vessels was harvested from a rat together with the blood vessels that supplied it, and the muscle flap was perfused via its artery and vein in the bioreactor chamber. Next, three cell sheets prepared separately by co-culture of cardiomyocytes and vascular endothelial cells were laminated onto the vascular bed, and perfusion culture was performed using solution supplemented with basic fibroblast growth factor. Perfusion culture for three days resulted in the myocardial tissue developing capillaries that connected to those of the vascular bed. In vitro construction of myocardial tissue with a thickness of approximately 0.2 mm was achieved by stacking three-layered cell sheets in a stepwise fashion four times. Furthermore, vascularized myocardial tissue prepared in this way was successfully transplanted into an animal by anastomosis of the graft’s artery and vein with the neck vessels in the host. Importantly, the myocardial graft tissue was still beating spontaneously two weeks after transplantation, demonstrating that our technique might be suitable for future use in therapeutic applications [35].

The research described above confirmed that functional capillaries could be introduced into bioengineered tissue in an ex vivo environment, raising the possibility that this method could be used to produce thick sections of functional artificial 3D tissue or even organs. In particular, it was shown that thicker sections of tissue could be created in vitro or in vivo by the stepwise stacking of cell sheets and that co-culture with vascular endothelial cells could enhance angiogenesis. The repeated lamination of two-dimensional cell sheets, which become connected to each other both physically and functionally via adhesion proteins, can be used to produce 3D tissues of high cell density without the need for artificial materials such as scaffolds. Furthermore, utilizing cell sheets containing a network of vascular endothelial cells together with a perfusable vascular bed promotes angiogenesis and overcomes some of the major limitations of conventional methods, enabling functional 3D tissues to be engineered that can be perfused in vivo by anastomosis of their blood vessels with those of the host. 

Pioneering work into the construction of scaffold-free tissues from cell sheets, promotion of angiogenesis, perfusion of tissues ex vivo, and transplantation of grafts with anastomosable blood vessels enabled us to engineer myocardial tissue that exhibits synchronized beating and possesses blood vessels that can be connected to those of a host. It is currently being developed for technologies with the aim of generating large-sized tissues for potential clinical use in patients.

With the above objective in mind, it was examined whether the porcine small intestine (a 3D structure with clinically relevant dimensions) could be used as a perfusable vascular bed in a bioreactor to construct human cardiac tissue with a vascular network in vitro. The vascular bed was made by harvesting a segment of the porcine small intestine (along with a branch of the superior mesenteric artery and a branch of the superior mesenteric vein), incising it longitudinally, and removing the mucosa. Cardiomyocyte sheets were generated by co-culturing human cardiomyocytes derived from human-induced pluripotent stem cells (hiPSCs) with endothelial cells and fibroblasts. Triple-layered cardiomyocyte sheets were placed on the vascular bed, and the resulting construct was perfused and cultured in a bioreactor. After perfusion culture for 24 h, the cardiac tissue exhibited spontaneous and synchronous beating and generated regular action potentials at a rate of 105 ± 13/min. Moreover, immunohistochemistry experiments revealed that some connections had formed between the blood vessels of the cardiac cell sheets and those of the vascular bed. Therefore, this technique allows us to use human cardiac cell sheets and porcine small intestine-derived vascular beds to generate spontaneously beating 3D human heart tissue in vitro that can be transplanted in vivo by the anastomosis of its blood vessels with those of the host. Further development of this method may lead to the creation of functional heart tissue that can be used to treat severe heart failure.

In addition, it is believed that the scalability and functionality of bioengineered 3D tissues can be improved through the use of in vitro systems that mimic the in vivo conditions as closely as possible. For example, there is interest in evaluating the possible benefits of the mechanical stimulation of the tissue during culture as well as the use of biochemical stimuli to enhance vascular system development. Therefore, bioreactor systems that incorporate completely new concepts in tissue perfusion culture were developed. 

One research area of recent interest is the optimization of perfusion conditions in order to better preserve donor organs and improve the culture of organs generated by tissue engineering. Therefore, it was evaluated whether intermittent pressurization of tissues (and their blood vessels) ex vivo would enhance the perfusion and survival of cultured organs. To achieve this, new chambers and bioreactor systems were developed to apply external pressure during the perfusion culture of rat small intestine and thigh muscle. Intermittent pressurization resulted in uniform perfusion of small intestinal specimens and minimal tissue damage after 20 h of perfusion, whereas non-pressurized specimens showed histological evidence of tissue damage to the upper surface. Longer-term studies in rat thigh muscle specimens showed that intermittent pressurization resulted in better perfusion throughout the 14-day experimental period as well as improved organ survival, greater preservation of vascular structures and skeletal muscle nuclei, and reduced levels of tissue necrosis. Thus, the intermittent application of external pressure prolonged the survival of the small intestine and skeletal muscles by enhancing their perfusion [36]. We believe that this innovative perfusion technique could help to improve the preservation of donor organs and culture of bioengineered organs.

## 3. Future Perspectives

The development of organ-like tissues is the next important challenge facing regenerative medicine. Organ-like structures are already being synthesized on a small scale, but overcoming current problems with cell sourcing and upscaling will enable the engineering of cardiac assist devices and organs for transplantation.

A major obstacle to tissue engineering is the inadequate supply of oxygen to 3D structures, which limits the thickness of the structures to approximately 0.1 mm. Therefore, the establishment of 3D tissue construction techniques that incorporate capillaries and lymphatic vessels (to supply oxygen and nutrients and remove waste products) will expand the application of regenerative medicine and increase its sophistication. Techniques such as growth factor administration, gene transfer, and co-culture with vascular progenitor cells could be used to enhance the development of vascular networks and overcome the limitations of passive diffusion, thereby allowing the creation of thicker and stronger myocardial tissue. 

It has been demonstrated that the stepwise stacking of cell sheets can generate thick sections of tissue, that functional capillaries can be induced in these bioengineered tissues both in vivo and in vitro, and that angiogenesis can be enhanced by the inclusion of vascular endothelial cells in the cell sheets. Based on these technologies, we expect to be able to produce large pieces of tissue for potential clinical applications in the future. Further development of vascular beds and bioreactors for in vitro perfusion and vascularization of artificial tissues made from cell sheets will facilitate the ultimate goal of creating complex and highly functional organs that have high vascular requirements such as the heart, kidney, and liver.

## 4. Conclusions

In this review, we have described the current state-of-the-art tissue engineering techniques that can be used to construct 3D tissues and potentially functional organs. In particular, we have focused on methods for introducing vascular networks into myocardial tissue engineered from cell sheets. Cell sheets prepared using temperature-responsive culture dishes have several advantages over other currently available methods, including more efficient transplantation, the construction of tissues that cannot be made with existing techniques, and the ability to stratify cells. In addition, cell sheet-based therapies have shown promising results in human clinical trials.

The successful construction of functional tissues and organs will require many challenges to be overcome, including the development of cells that can be transplanted into humans, the generation of larger sized tissues, and the preservation of grafts before transplantation. Tissue engineering evolved from a collaboration between biology- and engineering-related fields, and we believe that further cooperation between multiple disciplines will deepen our understanding and lead to new discoveries. The development of technologies based on an interdisciplinary approach will make it possible to create new therapies for diseases that are difficult to treat at present.

## Figures and Tables

**Figure 1 ijms-22-00092-f001:**
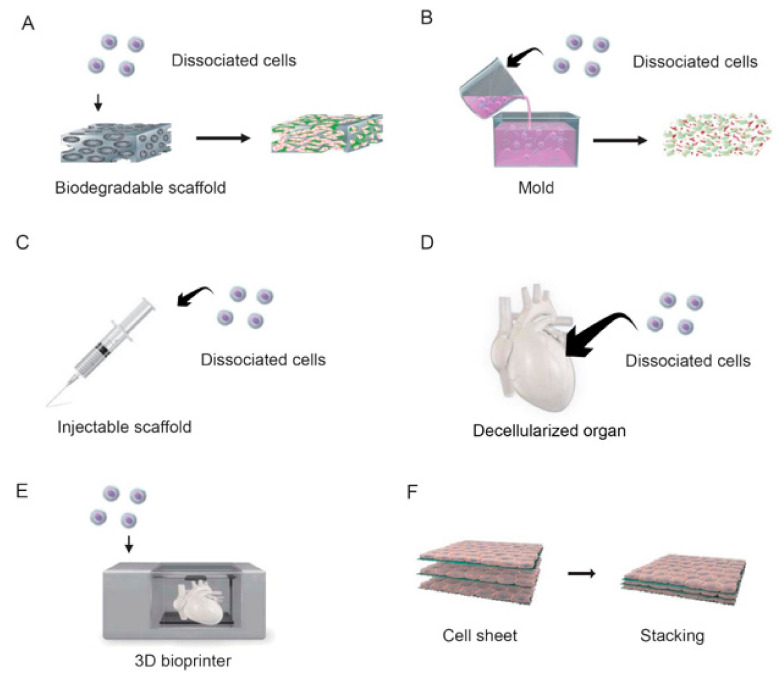
Approaches to myocardial tissue engineering. (**A**) Dissociated cells are poured into a pre-made, highly porous scaffold. The scaffold undergoes biodegradation, and extracellular matrix (ECM) fills the intercellular space to create a 3D tissue. (**B**) A mixture of dissociated cells and biodegradable polymers is poured into a mold, and the molecules are polymerized. Since the gel is poured into a mold, tissue of any shape can be constructed. (**C**) Cells suspended in a polymer solution are injected directly into injured/defective tissue using a syringe. This approach is similar to that used for cell infusion therapy. (**D**) Cells are actively removed from human or xenobiotic tissues and organs to leave only the ECM component, which serves as a scaffold for the seeding of dissociated cells. Since the 3D structure of the original tissue is preserved, the entire organ can be reconstructed. (**E**) A 3D printer uses a mixture of cells and polymer gel as bio-ink to assemble 3D tissues in a bottom-up fashion. (**F**) Cell sheets harvested from temperature-responsive culture surfaces are layered. The cell sheets adhere to each other via the ECM that they produce, yielding a 3D tissue that does not contain a biodegradable scaffold. The schematic diagram was drawn with the graphic design software, Adobe Illustrator.

**Figure 2 ijms-22-00092-f002:**
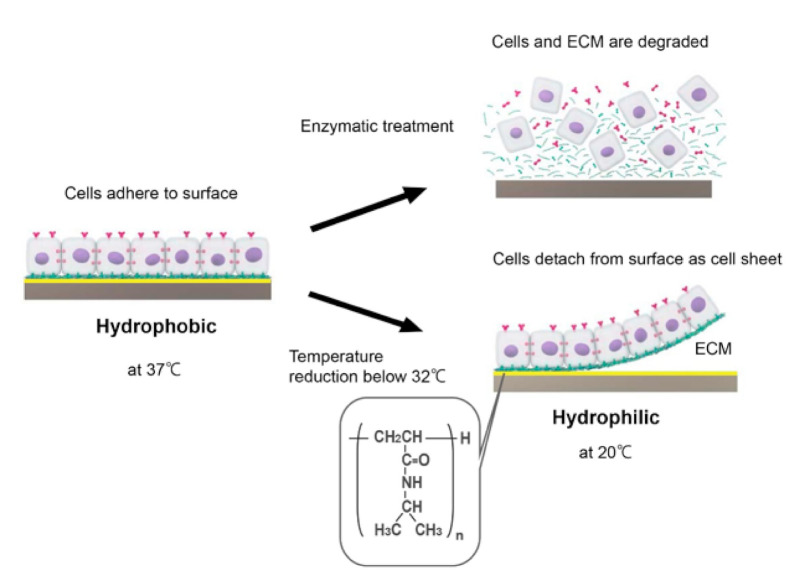
Generation of a cell sheet on a temperature-responsive surface. The temperature-responsive dish allows cultured cells to be harvested as an intact, confluent sheet simply by lowering the temperature (culture in a CO_2_ incubator set at 20 °C). This technique prevents the need for enzymatic harvesting. The schematic diagram was drawn with the graphic design software, Adobe Illustrator.

**Figure 3 ijms-22-00092-f003:**
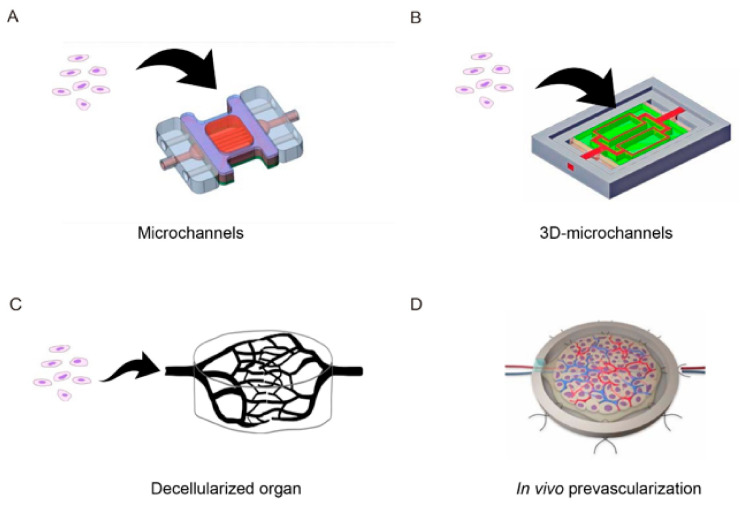
Technologies available for the induction of capillaries in bioengineered tissue. (**A**) A microfluidic system is used to introduce artificial capillaries (microchannels) within the tissues that are perfused with fluid. (**B**) A microfluidic system based on the creation of a scaffold that imitates a 3D microcirculatory structure. Vascular endothelial cells are seeded on the luminal side of the scaffold and perfusion culture is performed. (**C**) A whole organ is decellularized and vascular endothelial cells are seeded into the original vascular structure and subjected to perfusion culture so as to reproduce a circulatory system. (**D**) Engineered tissue is implanted onto well-vascularized host tissue to induce the growth of capillaries into the graft. The host tissue is supplied by blood vessels that are suitable for anastomosis onto other blood vessels, enabling later transplantation of the vascularized graft to the desired site. The schematic diagram was drawn with the graphic design software, Adobe Illustrator.

**Figure 4 ijms-22-00092-f004:**
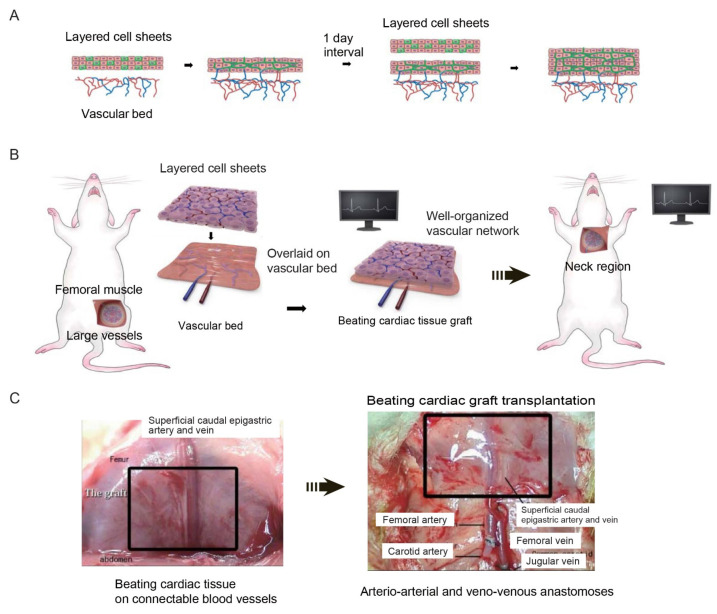
Construction of thick cardiac tissue grafts with blood vessels by the sequential implantation of multiple cardiomyocyte sheets containing vascular endothelial cells. (**A**) A three-layered cardiac cell sheet containing endothelial cells is transplanted onto a vascular bed. One day later, after a capillary network has formed in the cardiac tissue graft, a new three-layered cardiac cell sheet containing endothelial cells is transplanted on top of the original cardiac cell sheet. Thick myocardial tissue with a capillary network is constructed by repeating this procedure multiple times. (**B**) Fabrication of cardiac tissue suitable for ectopic transplantation. Three-layered cardiomyocyte sheets containing endothelial cells are repeatedly grafted onto existing large vessels in vivo to create pulsatile cardiac tissue supplied by an artery and vein. (**C**) A vascularized cardiac tissue graft is fabricated by the repeated implantation of three-layered cardiomyocyte sheets onto the femoral muscle of a rat. The cardiac tissue graft and accompanying femoral blood vessels are then excised and implanted into a region of the neck by anastomosis of the femoral blood vessels with neck vessels. Pulsation of the graft resumes immediately after vascular anastomosis. Reproduced with permission, Copyright 2006 Federation of American Societies for Experimental Biology. The schematic diagram was drawn with the graphic design software, Adobe Illustrator.

**Figure 5 ijms-22-00092-f005:**
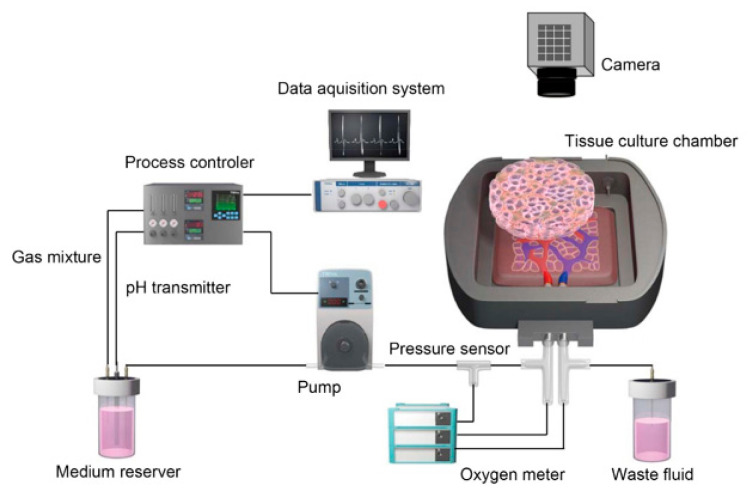
Tissue perfusion culture system and stacking of cell sheets on a vascular bed. The schematic diagram was drawn with the graphic design software, Adobe Illustrator.

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
