# Peer review of "Capillary Networks for Bio-Artificial Three-Dimensional Tissues Fabricated Using Cell Sheet Based Tissue Engineering"

_ijms, 2020, doi:10.3390/ijms22010092_

Round 1

Reviewer 1 Report

            The authors of the manuscript “Capillary networks for bio-artificial three-dimensional tissues fabricated using cell sheet engineering” are presenting an interesting review. The aim of the review are techniques leading to formation of a 3D capillary network.

The topic is undoubtedly very interesting and encouraging. The manuscript, however, suffers from several major shortcomings hindering publication in the present form.

Major comments

  1. The authors declare their manuscript as a review. According to my opinion, however, the authors are trying to convince readers about advantages of their approach. There are many other missing approaches and publications regarding the review topic. I would recommend the authors rewriting the manuscript to cover the topic in full. Alternatively, the authors should reconsider the review title and make it more specific.
  2. The authors emphasize the role of bulk chemical factors but they almost completely omit physical structures and targeting, like application of functionalized nanofibers and nanostructures, namely functionalized with suitable growth factors or other bioactive substances for oriented cell migration and angiogenesis.

Minor comments

  1. I also recommend rewriting the abstract. The authors are presenting the review, not an advertisement for their work and results.

Author Response

Comments from Reviewer #1 

The authors of the manuscript “Capillary networks for bio-artificial three-dimensional tissues fabricated using cell sheet engineering” are presenting an interesting review. The aim of the review are techniques leading to formation of a 3D capillary network.

The topic is undoubtedly very interesting and encouraging. The manuscript, however, suffers from several major shortcomings hindering publication in the present form.

We thank the reviewer for their constructive comments regarding our manuscript.

The authors declare their manuscript as a review. According to my opinion, however, the authors are trying to convince readers about advantages of their approach. There are many other missing approaches and publications regarding the review topic. I would recommend the authors rewriting the manuscript to cover the topic in full. Alternatively, the authors should reconsider the review title and make it more specific. The authors emphasize the role of bulk chemical factors but they almost completely omit physical structures and targeting, like application of functionalized nanofibers and nanostructures, namely functionalized with suitable growth factors or other bioactive substances for oriented cell migration and angiogenesis.

Response

We thank the reviewer for your helpful recommendation. The use of growth factors for angiogenesis have already mentioned briefly in the in vitro approach section of overview.

I also recommend rewriting the abstract. The authors are presenting the review, not an advertisement for their work and results.

Response

We appreciate the reviewer’s remarks on this point. However, we believe the original text is correct, as this manuscript was invited by a guest editor to write our original technology for a special issue of Thinking in 3D: From Molecules to Organisms.

Reviewer 2 Report

One of the most important and intractable challenges of tissue engineering is the formation of a sufficient vascular network when creating or regenerating structures of tissues and organs. Conventional tissue engineering technologiesrely on simple diffusion to deliver oxygen/nutrients and remove waste products. Tissues generated with these approaches are limited in terms of their thickness and functionality. The construction of tissues with higher vascular requirements (such as heart) required the development of innovative techniques to achieve functional vascularization of the bioengineered graft. This review describes some of the methods that can be used to construct vascular networks within bioengineered three-dimensional organs, with a focus mainly on their own research into the generation of myocardial tissue. Authors developed a method of creating thick, functional myocardial tissue based on the transplantation of multiple cell sheets onto host tissue in a stepwise manner, leaving sufficient intervals between transplantations to allow angiogenesis to generate capillaries within the graft.

The review is framed correctly. The authors used the original understandable Figures, but did not indicate the graphics program used. The authors cite sufficient original articles, mainly their own. The authors can correct the title and make accent to the fact in the abstract that the review was written mainly on their own experience. The previously received results have been published in high IF journals. The described models are adequate and can be reproduced by other researchers. The authors have a clear concise writing style. The review will be of interest to researchers who study optimization of perfusion conditions in order to better preserve donor organs and improve the culture of organs generated by tissue engineering.

I recommend the article for publication in the journal «IJMS» after making the recommended changes.

Author Response

Comments from Reviewer #2

One of the most important and intractable challenges of tissue engineering is the formation of a sufficient vascular network when creating or regenerating structures of tissues and organs. Conventional tissue engineering technology on simple diffusion to deliver oxygen/nutrients and remove waste products. Tissues generated with these approaches are limited in terms of their thickness and functionality. The construction of tissues with higher vascular requirements (such as heart) required the development of innovative techniques to achieve functional vascularization of the bioengineered graft. This review describes some of the methods that can be used to construct vascular networks within bioengineered three-dimensional organs, with a focus mainly on their own research into the generation of myocardial tissue. Authors developed a method of creating thick, functional myocardial tissue based on the transplantation of multiple cell sheets onto host tissue in a stepwise manner, leaving sufficient intervals between transplantations to allow angiogenesis to generate capillaries within the graft.

The review is framed correctly. The authors used the original understandable Figures, but did not indicate the graphics program used. The authors cite sufficient original articles, mainly their own. The authors can correct the title and make accent to the fact in the abstract that the review was written mainly on their own experience. The previously received results have been published in high IF journals. The described models are adequate and can be reproduced by other researchers. The authors have a clear concise writing style. The review will be of interest to researchers who study optimization of perfusion conditions in order to better preserve donor organs and improve the culture of organs generated by tissue engineering.

Response

We greatly appreciate the reviewer’s helpful and important comments and positive feedback regarding our manuscript.

We have now changed the title from “Capillary networks for bio-artificial three-dimensional tissues fabricated using cell sheet engineering” to “Capillary networks for bio-artificial three-dimensional tissues fabricated using cell sheet based tissue engineering to make the title more specific.

Accordingly, at the end of each figure legend we have added the following explanation. “The schematic diagram was drawn by graphic design software, adobe illustrator”.